# Prevalence and Characteristics of Back Pain in Children and Adolescents from the Region of Murcia (Spain): ISQUIOS Programme

**DOI:** 10.3390/ijerph19020946

**Published:** 2022-01-15

**Authors:** María Teresa Martínez-Romero, Antonio Cejudo, Pilar Sainz de Baranda

**Affiliations:** 1Department of Physical Activity and Sport, Faculty of Sport Sciences, Regional Campus of International Excellence “Campus Mare Nostrum”, University of Murcia, 30720 Murcia, Spain; mariateresa.martinez13@um.es; 2Sports and Musculoskeletal System Research Group (RAQUIS), Regional Campus of International Excellence “Campus Mare Nostrum”, University of Murcia, 30720 Murcia, Spain

**Keywords:** low back pain, mid-back pain, neck pain, back health, maturational stage, school children

## Abstract

Puberty is a vulnerable period for musculoskeletal disorders due to the existence of a wide inter-individual variation in growth and development. The main objective of the present study was to describe the prevalence of back pain (BP) in the past year and month in school-aged children according to sex, age, maturity status, body mass index (BMI) and pain characteristics. This study involved 513 students aged between 9 and 16 years. Anthropometric measures were recorded to calculate the maturity stage of the students using a regression equation comprising measures for age, body mass, body height, sitting height and leg length. An ad hoc questionnaire composed of eight questions was used to describe BP prevalence in school-aged children. The results showed that the prevalence of BP in school-aged children was observed in 35.1% over the last year (45% boys and 55% girls), and 17.3% (40.4% boys and 59.6% girls, with an association found between female sex and BP) in the last month. The prevalence of back pain in the past year and month was higher the older the students were, or the more pubertal development they had experienced. The prevalence of BP in the last year was also higher in those with overweight or obesity. After adjustment for sex, there was an association between BP and older age and higher BMI in boys and an association between BP and higher pubertal development in girls. In summary, the present study showed that the prevalence of BP was related to the maturity stage and weight of the participants, with different prevalence patterns found according to sex.

## 1. Introduction

Nowadays, back pain (BP) in childhood and adolescence is no longer considered to be uncommon or rare [1,2]. Most studies focus on investigating the prevalence of low back pain (LBP), as it is most common in children and adolescents [3]. However, pain in the neck and thoracic region (mid-back) should not be underestimated, as the prevalence increases during this growth phase [4,5].

Aartun et al. [6] found that neck pain (NP), mid-back pain (MBP) and low back pain (LBP) were most prevalent at 11–13 years of age, and at follow-up (two years later), this prevalence had increased for each back region. In terms of the progression of pain, they concluded that pain appeared to be mild in nature, relatively infrequent and of low intensity in the majority of participants, while a group of participants (14–20% of the sample) were more frequently and severely affected by BP. In a Spanish sample, the prevalence of BP within the last week in school children (8–12 years) was reported to be 10.6% (1.7% NP, 7.7 MBP and 2.9% LBP) and in high school students (12–17 years) the prevalence of BP within the last week was 25.7% [7,8]. The presence of BP in childhood and adolescence can lead to absenteeism from school, the need for medical treatment and medication [1,9] and limitations to activities of daily living (ADL) (e.g., standing in a queue, carrying a backpack or performing physical activities) [1,2].

Considering that the prevalence of BP increases from childhood to adolescence, especially at the age of 11–12 years, or even more after the onset of puberty [10,11,12]. The puberty (a transitional period, characterized by rapid physiological and anatomical changes) seems to be the time when the prevalence levels of BP increase rapidly, with this prevalence being higher in girls due to their early onset of pubertal development (which is, on average, 2 years earlier than boys) [3,11,13]. A recent 3-year prospective study of spinal pain in children (9 years) concluded that both pubertal development and linear growth are associated with spinal pain. It was observed that the frequency and duration of spinal pain increased in boys and girls with advanced pubertal development and greater growth [14]. 

Currently, epidemiological studies are still refer to the school year or age of children and adolescents, although it is known that there is a link between puberty and the development of BP. The timing and pace of the maturation process varies greatly from person to person, leading to differences in size, shape and body composition in children and adolescents of similar ages [15,16,17]. At a given age, adolescents of the same sex may be at very different stages of maturation; for some, pubertal development has not yet begun, while for others it is already complete. Therefore, biological age is the reference point for working with young people [16].

Previous studies have shown that the musculoskeletal structures of children at peak height velocity (PHV) are not mature enough to support the sudden mechanical load changes in the spine caused by the difference in growth rates between the legs and trunk, with the long bones of the legs experiencing a growth peak before the shorter bones of the trunk [18,19]. The development of muscles and the adaptation of ligaments and tendons to these new stresses also do not occur at the same rate. Therefore, the growth spurt phase could be considered to be a critical period compared to the episodes before or after this phase [11,14,20,21]. 

These structural changes, together with physical (increasing height and change in body composition) and hormonal changes (females secrete more estrogen and males more testosterone) [13,19,22] lead to an increase in height and muscle mass in boys and an increase in body fat in girls during the onset of puberty [23,24,25]. Weight gain during puberty in girls has been associated with the increased accumulation of fat mass, possibly related to low motor skills and low participation in physical activities [15,17]. It has been suggested that overweight and obesity may influence the occurrence of BP [1,26,27]. In addition, the mechanical load on the spine may increase in people with obesity due to the increased compressive force on the structures of the lumbar spine [26]. In this context, childhood obesity may be a modifiable risk factor for back pain, and weight optimization offers benefits beyond musculoskeletal health [1,27]. Puberty is a vulnerable time for musculoskeletal disorders because of the large interindividual differences in growth and development. With this in mind, previous studies [7,8] have not examined variables, such as sex, age, stage of maturity, BMI, affected back region and pain BP (frequency, intensity and severity), that might influence the prevalence of BP in the last year or month in school children. Therefore, the objectives of this study were a) to describe the prevalence of BP in the last year or month in a sample of school-aged children from the Region of Murcia (Spain) in relation to sex, age, stage of maturity, BMI, affected back region and the frequency, intensity and severity of pain, and b) to determine the possible factors associated with this condition. We hypothesise that the prevalence of BP increases with the progression of puberty, as does the weight of school children, in both boys and girls.

## 2. Materials and Methods

### 2.1. Study Design 

A cross-sectional study was conducted to determine the prevalence of BP in children and adolescents of different sex, age and maturity status. All measurements were taken before participation in a posture and fitness programme called the “ISQUIOS Programme”. The study was conducted during the first semester of the 2017–2018 and 2018–2019 school years.

### 2.2. Participants 

More than 50 educational centres in the Region of Murcia were initially invited to participate voluntarily in this study. These educational centres participated in the ISQUIOS Programme, an educational programme for posture education. The physical education teachers of each educational centre decided whether they wanted to participate in the study or not.

The exclusion criteria were: (a) a diagnosed spinal condition or major physical injury (reported by the participants), (b) no return of the signed consent form (from both parents/guardians and students) before the start of the study, and (c) failure to complete the questionnaire on BP.

A full oral description of the nature and purpose of the study was given to the students and their parents/guardians and the physical education teachers. The protocol was fully approved by the Review Committee for Research Review Board at the University of Murcia (Spain) (ID: 1920/2018) and in accordance with the Helsinki Declaration.

Finally, a sample of 513 students from 10 primary schools and 2 secondary schools aged 9 to 16 years (49.9% female; mean age ± SD = 12.6 ± 1.9 years) participated in this study. One hundred and fifteen students were excluded due to the exclusion criteria (Figure 1).

### 2.3. Procedure

Participants were tested in two separate sessions during physical education classes. In the first session, anthropometric measurements were taken to determine the maturity stage of the students. In the second session, an experienced researcher presented the questionnaire to the older students (secondary school) and parents (in the case of primary school students), explained the procedure for completing the survey and personally answered all the participants’ questions. The secondary students completed the questionnaire during physical education classes, while the primary students completed the BP questionnaire with their parents at home. The use of parental reports is very important for children of early school age (under 11 years) as it helps to improve the quality of the information collected and has been recommended in previous studies [28].

### 2.4. Anthropometry and Maturity

Body weight in kilograms was measured on a calibrated physician scale (SECA 799, Hamburg, Germany). Body height was recorded in centimetres on a measuring plate (SECA 799). Seat height was measured in centimetres. Leg length was calculated as the difference between body height and seat height. Body mass index was determined by dividing body mass by height (in metres) squared. Classification was performed according to the age- and sex-specific percentile scale of the World Health Organization Z-score [29]: underweight (equal to or less than 3%), normal weight (from 3.01% to 84.99%), overweight (from 85% to 94.99%) and obesity (equal to or greater than 95%).

The stage of maturation was calculated in a non-invasive manner using a regression equation that incorporated the measurements of age, body mass, height, sitting height and leg length collected during the first part of the testing sessions [30]. This method was used to complete the maturity offset (calculation of years from PHV) (Equation (1) for boys and Equation (2) for girls). Due to the error in the prediction equation of approximately 6 months in the paediatric population [30], participants with a maturity offset of −0.99 to −0.51 years and +0.51 to +0.99 years were removed from the maturity analysis [31,32]. In addition, participants whose maturity offset was outside of −3 or +3 years were removed from the analysis of the maturational stage to maximise accuracy [31]. This approach allowed the identification of 3 different maturity groups: pre-PHV (maturity offset of less than −1), circa-PHV (maturity offset between −0.5 and +0.5), and post-PHV (maturity offset of more than +1). Therefore, the following equations were used to calculate the maturity difference:(boys) = −9.236 + 0.0002708 × (leg length × sitting height) − 0.001663 × (age × leg length) + 0.007216 × (age × sitting height) + 0.02292 × (weight ⁄ (height × 100))(1)
(girls) = −9.376 + 0.0001882 × (leg length × sitting height) + 0.0022 × (age × leg length) + 0.005841 × (age × sitting height) − 0.002658 × (age × weight) + 0.07693 × (weight ⁄ (height × 100))(2)

### 2.5. Back Pain Assessment

An ad hoc questionnaire [3,7,33,34,35] consisting of sociodemographic questions and 8 BP questions was used to describe the prevalence of BP in male and female school children (1-year and 1-month prevalence, spinal pain regions, frequency, intensity, sciatica and limitation of activities of daily living (ADL)). The reliability and validity of the questionnaire has been investigated in previous studies with Spanish children and adolescents [35,36]. The questionnaire starts with sociodemographic questions concerning sex, age, school, grade level and diagnosed pathologies. This is followed by questions on the prevalence of BP in the last 12 months and in the last 1 month (“Have you had back pain in the last year (or month)?”). Those who suffered from BP within the last year answered questions about the region of pain in the spine (neck, middle back and/or lower back) and the characteristics of the pain in the last year (frequency and intensity of pain, radiation to the lower limbs (sciatica) and limitations in activities of daily living). BP was defined as pain or discomfort in specific parts of the back that was not due to trauma or menstrual pain. 

The questionnaire included a drawing of the back to mark the most affected region (Figure 2).

The frequency of spinal pain was categorised as “at least once”, “a few days”, “frequently (several days)” or “daily”. The intensity of spinal pain was quantified using a visual analogue scale (VAS) from 0 (no pain) to 10 (unbearable pain).

Sciatica was assessed by asking whether the back pain had ever spread to a leg. To find out if the back pain had limited ADL, it was asked if the pain was severe enough to alter daily routines for more than one day.

### 2.6. Statistical Analyses

Data were analysed according to sex using descriptive statistics. 

Logistic regression analyses (forced entry/enter method), including the following categorical (binary) factors (coding 0/1), were performed to identify possible associations: sex (male/female), pre-PHV status (no/yes), circa-PHV status (no/yes), post-PHV status (no/yes), under/normal weight (no/yes), overweight/obesity (no/yes), ADL limitation (no/yes) and sciatica (no/yes). Frequency was evaluated as an ordinal variable. Age and VAS were evaluated as continuous variables.

Associations were determined 1) for back pain in general (per sex) and 2) for pain per spinal region (LBP, MBP, NP and pain in more than one spinal region). For the analysis per spinal region, participants with pain in the region of interest were compared to children without pain in any spinal region. 

Effect sizes for ORs were defined as follows: small effect OR = 1 to 1.25, medium effect OR = 1.25 to 2 and large effect OR ≥ 2. 

The independent *t*-test was used to examine differences between students with or without BP (last year and last month) and those who reported ADL limitations and pain intensity. 

To examine differences between pain intensity and pain location or frequency, a one-way ANOVA was performed with a post hoc Scheffé test.

The statistical analyses were performed using the statistical package SPSS v. 21.0 for Windows (IBM Corp., Armonk, NY, USA). The significance level was set at *p* < 0.05.

A post hoc sample size calculation was performed with the software package G*Power 3.1.9.7. (Heinrich Heine-Universität Düsseldorf, Düsseldorf, Germany) using a Mann–Whitney (means between two group) test.

## 3. Results

The statistical power of the sample was calculated retrospectively for the variables for which significant differences were found between the classification groups (back pain vs. asymptomatic) using the input parameters sample size (age, PHV, and BMI), alpha level *p* < 0.05, effect size (BP last year: minimum value Hedges’ g = −0.361; BP last month: minimum value Hedges’ g = −0.454) for a Mann–Whitney (means between two group) analysis test (G*Power version 3.1.9.7, Heinrich-Heine-Universität Düsseldorf, Düsseldorf, Germany). The statistical power was 0.99 for age, BMI and PHV (BP last year), and 01.00 for age and BMI (BP last month).

This study is based on a sample of 513 school children aged 9–16 years (12.6 ± 1.9 years); 257 students (50.1%) were male and 256 students (49.9%) were female. The characteristics of the participants for the total sample and by sex are shown in Table 1. 

Regarding the main characteristics, 232 students (49.8%) were in a state before the onset of puberty, 138 students (39.6%) were in the pubertal phase, and 96 (20.6%) had already passed puberty. In terms of BMI classification, 245 school children (47.8%) were overweight or obese and 268 school children (52.2%) were underweight or normal weight. 

### 3.1. Back Pain Prevalence in the Last Year and Last Month

BP was observed in 180 students (35.1%) last year, comprising 81 boys (45%) and 99 girls (55%). Regression analyses showed that an increase in age was associated with BP in girls (OR = 1.35, 95% CI: 1.18–1.56, *p* < 0.001), but did not appear to be significant in boys (Table 2). Among school-aged children, the 1-year prevalence of BP was 28.4% in the pre-PHV phase, 34.1% in the circa-PHV phase and 52.1% in the post-PHV phase. However, only girls showed a consistent association between BP and greater pubertal development, with an OR of 2.75 (95% CI: 1.56–4.85, *p* < 0.001). The prevalence of BP in the last year was higher in overweight or obese boys (38.8%), with an association between overweight or obese boys and BP (OR: 1.97, 95% CI: 1.15–3.36, *p* = 0.01). 

In the last 1 month, 89 students (17.3%) had BP, comprising 36 boys (40.5%) and 53 girls (59.5%). Regression analyses showed that female sex was a risk factor for BP in the last month in children and adolescents, with an OR of 1.61 (95% CI: 1.01–2.55, *p* = 0.05) (Table 3). BP in the last month was more common in older students. As with last year prevalence, BP in girls was associated with age (OR = 1.39, 95% CI: 1.19–1.63, *p* < 0.001). Among school children, the 1-month prevalence of BP was 12.1% in the pre-PHV phase, 14.5% in the circa-PHV phase and 34.4% in the post-PHV phase, with an association between BP and girls in the post-PHV phase (OR = 3.58, 95% CI: 1.90–6.76, *p* < 0.001).

### 3.2. Prevalence of Back Pain by Spinal Region in the Last Year

The spinal region where pain was most commonly reported was the lower back (59.4%), followed by the middle back (37.2%) and the neck (21.1%). In the past year, 13.9% of students reported pain in more than one spinal region. Depending on the gender of the participants, girls had a higher prevalence of back pain for all spinal regions (NP: 8.2%, MBP: 16.4% and LBP: 21.5%) than boys (NP: 6.6%, MBP: 9.7% and LBP: 20.23%), with a significant association found between female sex and mid-back pain (OR: 1.82, 95% CI: 1.07–3.09, *p* = 0.02).

The prevalence in the different spinal areas was higher with age (NP: from 3.9% to 11.9%, MBP: from 11.8% to 19.05%, LBP: from 18.4% to 40.5%), with a significant association between older age and mid-back pain (OR: 1.15, 95% CI: 1.01–1.31, *p* = 0.03) and low back pain (OR: 1.12, 95% CI: 1.01–1.25, *p* = 0.03). Regarding maturity status, prevalence was lower before PHV and highest after PHV (NP: from 6.1% to 12.5%, MBP: from 10.8% to 21.9%, LBP: from 18.5% to 28.1%). A significant association was found between post-PHV status and the prevalence of neck pain (OR: 2.15, 95% CI: 1.04–4.43, *p* = 0.03), and mid-back pain (OR: 2.26, 95% CI: 1.27–4.0, *p* = 0.004).

According to the BMI classification, the prevalence of low back pain was higher in overweight or obese participants (24.9%) than in their non-overweight counterparts (17.2%). The prevalence was similar in normal-weight and overweight subjects in the cervical (normal: 6%, overweight: 9%) and thoracic regions (normal: 13.8%, overweight: 12.2%).

Regression analyses adjusted for sex showed that age and post-PHV status were risk factors for neck pain in girls, with ORs of 1.34 (95% CI: 1.07–1.67, *p* = 0.01) and 3.35 (95% CI: 1.35–8.31, *p* = 0.006), respectively. Female sex (OR: 1.82, 95% CI: 1.07–3.09, *p* = 0.02), older age (OR: 1.15, 95% CI: 1.01–1.31, *p* = 0.03) or post-PHV status (OR: 2.26, 95% CI: 1.27–4, *p* = 0.004) were found to be risk factors for mid-back pain. In addition, mid-back pain was associated with post-PHV boys (OR =3.17, 95% CI: 1.14–8.81, *p* = 0.02). For low back pain, older age (OR: 1.12, 95% CI: 1.01–1.25, *p* = 0.003), post-PHV (OR: 1.65, 95% CI: 1–2.74, *p* = 0.05) and being classified as overweight/obese (OR: 1.6, 95% CI: 1.04–2.46, *p* = 0.03) significantly increased the risk of pain in this area. For low back pain in girls, an association was also found with older age (OR: 1.28, 95% CI: 1.10–1.49, *p* = 0.002) and post-PHV status (OR: 1.97, 95% CI: 1.04–3.71, *p* = 0.03). In boys, the only association found was between low back pain and classification as overweight/obese (OR: 1.89, 95% CI: 1.01–3.52, *p* = 0.04).

### 3.3. Prevalence of Back Pain by Spinal Region in the Last Month

The regions where spinal pain occurred last month showed practically the same trend as last year. The most frequently reported region for spinal pain in the past month was the lower back (58.4%), followed by the middle back (34.8%) and the neck region (23.6%). In the last month, 15.7% reported pain in more than one spinal region. Depending on the sex of the participants, girls had a higher prevalence of back pain (NP: 4.7%, MBP: 6.6% and LBP: 12.9%) than boys (NP: 3.5%, MBP: 5.4% and LBP: 7.4%), with a significant association found between female sex and low back pain (OR: 1.85, 95% CI: 1.02–3.35, *p* = 0.04).

The prevalence in the different regions of the spine was higher with age (NP: from 2.6% to 7.1%, MBP: from 5.3% to 11.9%, LBP: from 5.3% to 21.4%), with a significant association between older age and low back pain (OR: 1.31, 95% CI: 1.13–1.51, *p* < 0.001). Regarding maturity status, the prevalence of pain was lower before PHV and highest after PHV. A significant association was found between post-PHV status and the prevalence of mid-back pain (OR: 2.57, 95% CI: 1.18–5.56, *p* = 0.01) and low back pain (OR: 3.16, 95% CI: 1.72–5.83, *p* < 0.001).

According to the BMI classification, the prevalence of low back pain was higher in overweight or obese participants (12.7%) than in their normal-weight counterparts (7.8%). 

Regression analyses adjusted for sex showed that age was a risk factor for neck pain in girls, with an OR of 1.37 (95% CI: 1.03–1.82, *p* = 0.03). Regarding low back pain, older age (OR: 1.46, 95% CI: 1.21–1.76, *p* < 0.001) and post-PHV (OR: 3.49, 95% CI: 1.65–7.39, *p* = 0.001) significantly increased the risk of low back pain in girls.

### 3.4. Characteristics of Back Pain

Almost half of the participants (48.9%) reported having had BP “a few days” in the past year. Of the 180 students with BP, only 22 (12.3%) had pain that prevented them from performing activities of daily living (ADL), and 11 (6.1%) reported suffering from sciatica. The positions in which BP was most common were standing (38.3%), bending the trunk forward (31.7%), sitting (30%) and lying down (8.3%). The mean intensity of BP was 3.87 ± 2.01 on the VAS. Non-recurrent pain (at least once) was rated on average as mild (2.7 ± 1.7 VAS points), occasional pain (a few days) (4.3 ± 1.8 VAS points) and frequent pain (often) (5.6 ± 1.4 VAS points) as moderate, while daily pain was rated as severe (6.3 ± 2.1 VAS points). 

Students who experienced BP in the past year and reported a frequency of “a few days” or “daily” showed an association with limitations in ADL (OR: 3.34, 95% CI: 1.76–6.34, *p* < 0.001). In addition, adolescents who reported limitations in ADL due to their spinal pain reported significantly higher pain intensity (5.1 ± 2.1 VAS points) than those whose spinal pain did not affect daily living (3.7 ± 1.9 VAS points) (*p* = 0.001). Differences were also found between the frequency and intensity of pain, with the intensity of BP being higher in those who reported a greater frequency of episodes (F_3,178_ = 20.38, *p* < 0.001). 

When BP characteristics were analysed in relation to sex, age, maturity status or BMI classification, associations were only found between higher pain frequency and older age (OR: 1.49, 95% CI: 1.19–1.89, *p* = 0.001), post-PHV status (OR: 4.21, 95% CI: 1.66–10.67, *p* = 0.002) or higher weight (OR: 3.19, 95% CI: 1.13–8.98, *p* = 0.03). On the other hand, there was an association between ADL limitations being presented and overweight/obesity status (OR: 6.83, 95% CI: 1.94–24.02, *p* = 0.003). 

Table 4 presents data on the characteristics of back pain in the past year, broken down by region of spinal pain. An association was found between participants with neck pain and higher pain frequency. Low back pain led to more limitations in daily life (LBP: 17.9%, pain in more than one spinal region: 16%, NP: 10.8%, MBP: 6%), with an association between low back pain and existing limitations. The intensity of low back pain (3.67 ± 2.18 VAS points) and neck pain (4.03 ± 1.65 VAS points) was lowest on average, followed by pain in more than one spinal region (4.25 ± 1.75 VAS points) and mid-back pain (4.26 ± 1.75 VAS points), with an association between mid-back pain and reporting a higher intensity of pain.

## 4. Discussion

The aim of this study was to describe the prevalence of BP and its characteristics in a sample of children and adolescents from the Murcia region, and to investigate the possible associations between the participants’ characteristics and BP.

### 4.1. Prevalence of Back Pain in the Last Year and Month

More than one third of participants (35.1%) reported having experienced BP in the past year, with similar prevalence in boys (45%) and girls (55%). BP was more common in older (post-PHV) (OR: 2.73, 95% CI: 1.67–4.47) and overweight or obese (OR: 1.58, 95% CI: 1.09–2.28) participants. The prevalence of BP in the last month showed the same trend for age and maturity level, albeit with a lower prevalence (17%), but not with BMI classification. However, an association was found between girls and BP (OR: 1.6, 95% CI: 1.01–2.55).

The mean 1-year and 1-month prevalence of BP found in the present study has lower values than those found in other studies. For example, Kedra et al. [37] reported a mean prevalence of BP of 76.2% among Polish students in the previous year (10–19 years). Aparicio-Sarmiento et al. [7] showed a mean 1-year prevalence of BP of 55.1% in Spanish high school students (12–17 years), although the mean 1-year prevalence in Spanish primary school students (8–12 years) was lower at 22.3%. Regarding prevalence in the last month, the mean prevalence of BP in a Swiss sample aged 10–16 years was 44.4% [38] and 33% in 9-year-old Danish children [39]. These differences in mean prevalence between studies could be due to the design and methodological quality of the studies, discrepancies in the definition of BP, the delineation of pain regions and the different instruments used to diagnose BP, the different survey periods and the age range and characteristics of the participants [2,3].

When the 1-year prevalence of BP was analysed by sex, the results showed different patterns. On the one hand, BP in boys was associated with a higher BMI (overweight/obesity), while in girls it was associated with completed pubertal development (post-PHV) and older age. In the sex-adjusted prevalence of BP in the last month, girls showed the same pattern as the occurrence of back pain in the previous year (age and post-PHV). For boys, however, no association was found. The results of the present study partly confirm the hypothesis made earlier for, when analysing the sex-adjusted prevalence, the hypothesis is confirmed by the fact that the higher the maturity level and age of the girls and the higher the weight of the boys, the higher the prevalence.

Sex-related differences in the prevalence of BP have always been present, with most studies finding a higher risk of BP in females [40]. Considering that BP in girls has been associated with older age and higher maturity status (post-PHV), it could be argued that back problems develop at puberty, or at least are more pronounced than in early childhood, as has been suggested by others [11]. 

The results of the present study are consistent with those found by Wedderkopp et al. [41] in a study of BP during the last month in girls (8–10 years and 14–16 years). They found an increase in BP during pubertal stages 3 and 4 (Tanner classification) because accelerated growth occurs during these stages and the back may be more vulnerable to mechanical injury. However, it is difficult to compare the results of this study with others because the few studies that have investigated the relationship between BP and pubertal development have used the classification proposed by Tanner [14,41] or the Pubertal Development Scale [10,16,42]. In our case, we chose the regression equation proposed by Mirwald et al. [30] because it is a non-invasive method for calculating the maturity offset. To our knowledge, there are several studies on BP and spinal development that use the maturity offset, but only with a pre-pubertal sample [43,44,45].

Another explanation could be that girls mature earlier than boys and that the hormonal changes that occur in the process alter their pain modulation and make them more susceptible to pain conditions [26,40,46]. In addition to these aspects, during pubertal development, children’s musculoskeletal structures are not mature enough to support the loads generated by growth, and this is also a critical period when most scoliosis or asymmetries of the body axis occur, which may contribute to the occurrence or increase in BP [18,46]. However, further research is needed to determine what circumstances during the onset of puberty may lead to an increase in reports of BP in girls.

In contrast, no association was found between BP and the stage of maturity of boys, but rather between BP and being overweight or obese. These findings are consistent with a recent longitudinal study that examined, among other things, the association between BP and childhood weight status (aged 4 to 15 years). Individuals who were classified as overweight or obese had a higher risk of developing BP at any site. In addition, after adjusting for age, sex, socioeconomic level and nationality, an obese child had a 34% higher risk of developing BP before the age of 15 [1]. 

### 4.2. Prevalence of Back Pain by Spinal Region in the Past Year and Month

The spinal region most frequently reported by BP (1-year and 1-month prevalence) was the lower back, followed by the middle back and neck. As with general BP, prevalence in the different spinal regions increased with age, pubertal development and BMI classification, but these associations occurred only in LBP.

In other studies, NP and MBP were also relatively low and showed a lower association with age. LBP was the most commonly reported site where the greatest increase with age was observed [39,41,43,47]. However, other authors noted that MBP was more frequently reported in young children (8–12 years) [8,39,41], while LBP was more common from age 13 years. 

Palmer et al. [1] have shown that children who are overweight or obese are at increased risk of developing BP in the lumbar, thoracic and cervical spine, supporting the findings of the present study, where overweight or obese participants showed an association with LBP in the last year and month. As mentioned earlier, being overweight or obese could increase the mechanical load on the spine by increasing the compressive forces on the immature structures of the spine, which in turn affects the nutrition of the intervertebral discs and increases the risk of LBP [18,26,27].

The prevalence of NP, MBP and LBP was higher in girls. Consistent with other studies [26,27,47] describing female sex as a significant risk factor for LBP and MBP, this study found that girls were 1.82 times more likely than boys to have MBP in the past year and 1.85 times more likely to have LBP in the past month. Sex-adjusted pain localization showed an association for LBP in overweight or obese boys in the last year (OR: 1.89, 95% CI: 1.02–3.49) and with older girls in the last year (OR for age: 1.28, 95% CI: 1.10–1.49, *p* = 0.00; OR for post-PHV: 1.97, 95% CI: 1.04–3.71, *p* = 0.03) and in the last month (OR for age: 1.46, 95% CI: 1.21–1.76, *p* < 0.001; OR for post-PHV: 3.49, 95% CI: 1.65–7.39, *p* = 0.001). Previous studies have found similar associations between being a girl and having LBP. Some explanations could be the greater curvature of the lumbar spine that they have compared to boys of the same age [8,48], or the hormonal changes associated with pubertal development that could influence pain perception [26,40,46]. In a recent study [49], menarche was found to be a predictive demographic or anamnestic factor associated with LBP in gymnasts (6 to 18 years old). Another reason is that girls reported LBP as a premenstrual symptom, although the explanatory notes to the questionnaire insisted that this was not the case. On the other hand, it is probably a consequence of the roles imposed by upbringing and society that boys avoid showing symptoms or feelings of pain, while this is more accepted in girls [50,51].

### 4.3. Characteristics of Back Pain

Regarding the characteristics of pain, both sexes behaved similarly (no differences or associations were found between the variables and sex). The frequency with which they suffered BP was occasional (a few days) with a mean intensity of 3.87 ± 2.01 VAS (moderate). A higher frequency of BP episodes was associated with a greater intensity of pain and a limitation of ADL, and adolescents who reported limitations of ADL due to their spinal pain reported significantly higher pain intensity. The study by Aartun et al. [6] found that average pain intensity was lower in students who reported pain “once or twice” and increased progressively and significantly in the “sometimes” and “often” groups. Similarly, Wirth and Humphreys [38] showed that about a quarter of adolescents with BP reported frequent pain of moderate or severe intensity. The same authors point out that regardless of the location of the pain, the increase in pain intensity and frequency is a predictor of whether the experience of BP had an impact on the adolescents’ daily lives [38].

On the other hand, a higher frequency was observed when participants were older (post-PHV) or overweight/obese. The increased frequency of BP with increasing age or advanced adolescent pubertal development is consistent with previous studies [6,14,37,38,39]. No differences were found between pain intensity and BP regions, but an association was found between LBP and ADL limitations, although pain intensity was the lowest.

Some limitations of the study should be pointed out. First, it is a cross-sectional study that only shows the association between possible risk factors for BP, so it is not possible to prove a cause–effect relationship. Another limitation is the period of coverage studied. The use of 1-year prevalence could lead to a large loss of information, as it depends on the ability of participants to remember pain. Therefore, for future follow-ups, lifetime prevalence will be used together with 1-month or 1-week prevalence, as well as asking about the characteristics of pain for 1-month or 1-week prevalence. On the other hand, when analysing the different spinal regions by sex for age, maturity stage and BMI classification, the number of participants in each group is reduced, so the OR could be inflated. However, as the trend of the results is the same as in the unadjusted analysis of spine regions, the association is considered to hold. In addition, sex hormone levels or information on menarche were not measured, which could influence the relationship between the factors. Finally, due to the fact that the sample is from the same Spanish region, the results cannot be extrapolated to other regions of Spain or other countries, as they are influenced by environmental, social, cultural and genetic factors specific to each region. However, the results of the study could be used within the community to promote prevention programmes in educational centres.

## 5. Conclusions

The present study showed that the prevalence of BP was associated with the maturational state and weight of the participants, with different prevalence patterns found after adjusting for sex. BP, as with LBP, was associated with overweight or obesity in boys, while it was related to greater pubertal development in girls. The characteristics of BP were also associated with weight and maturity status regardless of sex, with a higher frequency and limitations of ADL found in older or overweight/obese participants. Therefore, the present study supports the statement that the prevalence of BP should be reported by spinal region, sex and maturity stage, and at least be supplemented by the characteristics, frequency and intensity to provide a complete overview of the development and severity of BP in the young population.

Body weight is a modifiable risk factor, the optimisation of which could have a positive impact on musculoskeletal health in school-aged children. Although sex, age and state of maturity are biological and non-modifiable factors, they all need to be considered when implementing educational prevention programmes. In general, intervention programmes should begin at an early age (primary school age) or coinciding with the pubertal growth spurt and continue after the PHV (secondary school age). These programmes should include core strengthen exercises as well as mobility exercises for the hip muscles and some aerobic exercise for weight control. For more specific programmes, it would be necessary to differentiate and individualise according to the sex of the school children, as it has been confirmed that back pain affects boys and girls differently.

## Figures and Tables

**Figure 1 ijerph-19-00946-f001:**
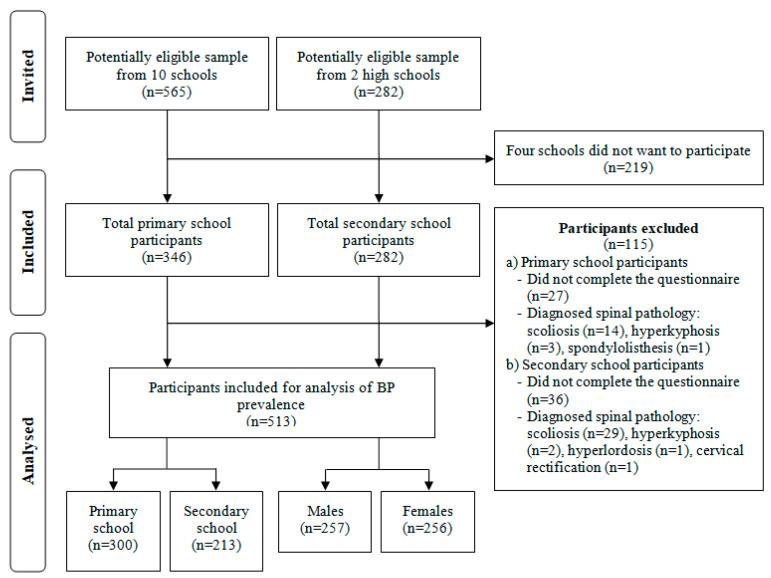
Flow diagram for the sample selection.

**Figure 2 ijerph-19-00946-f002:**
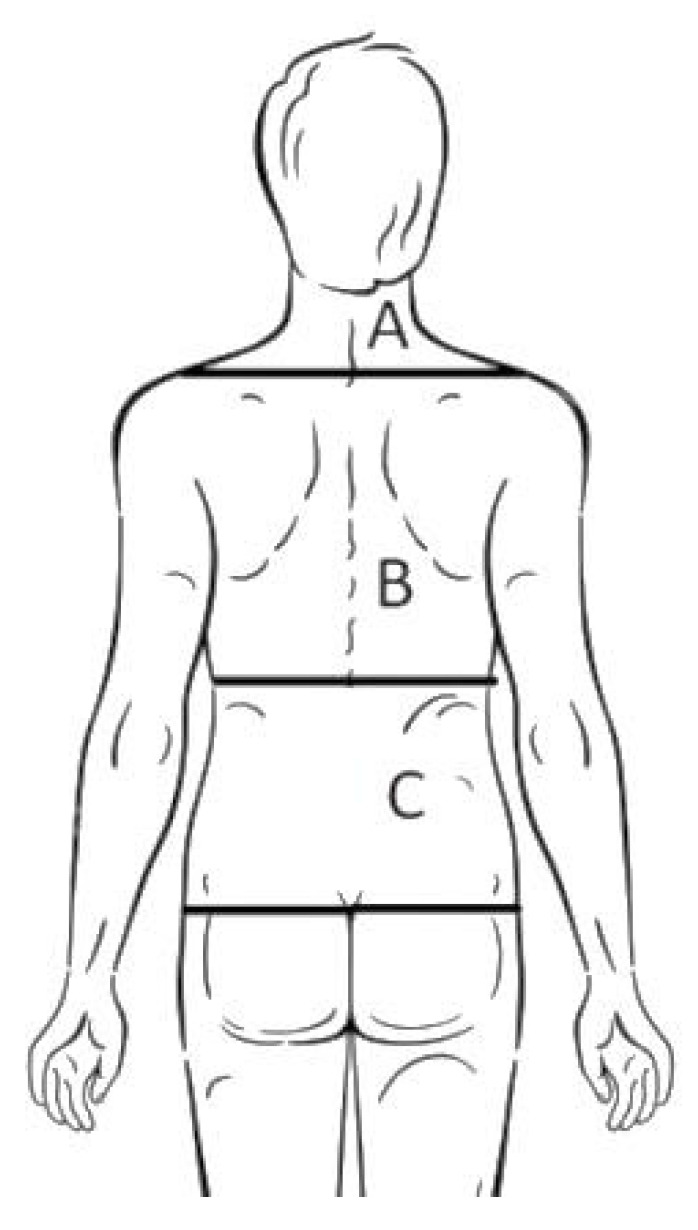
Drawing of the back to mark the region of back pain. **A**: neck pain, **B**: mid-back pain, **C**: low back pain.

**Table 1 ijerph-19-00946-t001:** Participants’ characteristics (mean ± SD) for the total sample and by sex.

	Total (N = 513)	Male (N = 257)	Female (N = 256)
Age (years)	12.06 ± 1.93	12.16 ± 1.93	11.96 ± 1.94
**Maturity status**			
Pre-PHV	−2.02 ± 0.58	−2.19 ± 0.56	–1.76 ± 0.52
Circa-PHV	0.002 ± 0.38	0.04 ± 0.44	–0.02 ± 0.34
Post-PHV	1.80 ± 0.56	1.43 ± 0.41	1.95 ± 0.54
**BMI classification**			
Underweight	1.71 ± 0.99	1.67 ± 1.0	1.80 ± 1.09
Normal weight	41.24 ± 21.82	41.27 ± 22.61	41.22 ± 21.15
Overweight	90.09 ± 4.01	90 ± 4.04	90.17 ± 4.01
Obese	98.15 ± 0.99	98.14 ± 0.99	98.16 ± 0.99

PHV: peak height velocity, BMI: body mass index.

**Table 2 ijerph-19-00946-t002:** Back pain in the last year and possible risk factors for the total sample and by sex.

BP Last Year	Total (N = 513, N with BP = 180)		Male (N = 257, N with BP = 81)		Female (N = 256, N with BP = 99)	
	OR (95% CI)	*p*	OR (95% CI)	*p*	OR (95% CI)	*p*
Sex	1.37 (0.95–1.97)	0.09	-	-	-	-
Age	**1.20 (1.09–1.32)**	**<0.001**	1.07 (0.94–1.23)	0.32	**1.35 (1.18–1.56)**	**<** **0.001**
**Maturity status**						
Pre-PHV	**0.58 (0.40–0.84)**	**0.04**	0.88 (0.52–1.49)	0.64	**0.41 (0.23–0.71)**	**0.02**
Circa-PHV	0.94 (0.62–1.42)	0.77	1.01 (0.53–1.89)	0.99	0.84 (0.48–1.44)	0.52
Post-PHV	**2.38 (1.53–3.76)**	**<0.001**	1.57 (0.69–3.56)	0.28	**2.75 (1.56–4.85)**	**<0.001**
**BMI classification**						
Under/normal weight	0.73 (0.51–1.05)	0.09	**0.51 (0.29–0.87)**	**0.01**	1.01 (0.61–1.68)	0.99
Overweight/obese	1.36 (0.95–1.96)	0.09	**1.97 (1.15–3.36)**	**0.01**	0.99 (0.60–1.65)	0.99

BP: back pain, PHV: peak height velocity, BMI: body mass index.

**Table 3 ijerph-19-00946-t003:** Back pain in the last month and possible risk factors for the total sample and by sex.

BP Last Month	Total (N = 513, N with BP = 180)		Male (N = 257, N with BP = 81)		Female (N = 256, N with BP = 99)	
	OR (95% CI)	*p*	OR (95% CI)	*p*	OR (95% CI)	*p*
Sex	**1.61(1.01–2.55)**	**0.05**	-	-	-	-
Age	**1.29 (1.15–1.45)**	**<** **0.001**	1.19 (0.99–1.42)	0.06	**1.39 (1.19–1.63)**	**<** **0.001**
**Maturity status**						
Pre-PHV	**0.49 (0.30–0.81)**	**0.005**	0.90 (0.44–1.80)	0.75	**0.31 (0.14–0.66)**	**0.003**
Circa-PHV	0.75 (0.44–1.29)	0.30	0.83 (0.34–1.99)	0.67	0.64 (0.32–1.29)	0.21
Post-PHV	**3.38 (2.03–5.60)**	**<0.001**	2.43 (0.94–6.24)	0.06	**3.58 (1.90–6.76)**	**<0.001**
**BMI classification**						
Under/normal weight	0.70 (0.44–1.11)	0.13	0.57 (0.27–1.16)	0.12	0.79 (0.43–1.46)	0.46
Overweight/obese	1.42 (0.90–2.26)	0.13	1.77 (0.86–3.63)	0.12	1.25 (0.68–2.29)	0.46

BP: back pain, PHV: peak height velocity, BMI: body mass index.

**Table 4 ijerph-19-00946-t004:** Characteristics of back pain in the last year by spinal pain region.

	Neck Pain (n = 38)		Mid-Back Pain (n = 67)		Low Back Pain (n = 106)	
	OR (95% CI)	*p*	OR (95% CI)	*p*	OR (95% CI)	*p*
Frequency	**3.08 (** **1.2–7.89)**	**0.01**	0.76 (0.29–1.98)	0.57	0.66 (0.27–1.62)	0.36
ADL Limitation	0.84 (0.27–2.65)	0.77	0.33 (0.11–1.04)	0.06	**5.04 (1.43–17.71)**	**0.006**
Sciatica	1.49 (0.37–5.91)	0.57	1.44 (0.42–4.91)	0.56	1.89 (0.48–7.36)	0.35
VAS	1.05 (0.88–1.26)	0.58	**1.17 (1–1.36)**	**0.05**	0.89 (0.76–1.03)	0.18

ADL: activities of daily living, VAS: visual analogue scale.

## Data Availability

The data sets used and analysed during the current study are available from the first or corresponding author on reasonable request.

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
