# Peer review of "Prevalence and Characteristics of Back Pain in Children and Adolescents from the Region of Murcia (Spain): ISQUIOS Programme"

_ijerph, 2022, doi:10.3390/ijerph19020946_

Round 1

Reviewer 1 Report

Dear authors,

Thank you very much for submitting this study. The design provides an interesting approach to the causes of back pain at different levels according to age, and relations between them.

I would recommend to review the abstract so a to comply with the introduction, methods, results and discussion.

The English is comprehensive, but there are a few details that would benefit from revision from an advanced English Speaker. These are things such as: in the flow diagram, in the participants excluded, the phrases from the bullet points of "Not fill out the...", or in line 131 "The classification was according to the...", line 218 "increased with increasing..."

In results, between lines 245-247 associations are described, but no information on them provided.

The results are clearly explained and the evolution of the study is well developed.

Author Response

Dear Reviewer,

Thank you for your review and constructive comments. We have reviewed the manuscript and addressed all your remarks (see the responses below). We hope that you consider the reviewed version of the manuscript worthy for publication.

Kind regards,

The authors.

Reviewer 2 Report

This is a very interesting paper, describing low back pain prevalence in children and adolescents aged 9-16 in Muricia. Since sample size is large the external validity of the study should be addressed .I disagree that the results are only regional since back pain the maturity exist everywhere.

I think the authors should be more “brave” to give carefully some recommendations based on the current study. For example – interventions should start in an early age. Special programs aiming at boys should be adjusted especially for certain ages, etc.

There are some major comments the authors should address. I hope they will help to strengthen the manuscript.

Abstract

Authors should change all active verbs such as “increased” into passive since they are not following the same children. This should be rephrased into: “was higher in older children or with increasing age”.

Introduction

Line 36- Prevalence was higher or increased? Was it a follow-up study?

Line 39 – A small number refers to 14-20% - of how many?

Lines 39-42 – Authors should explain why they repeated the study if they already have results in the same population.

Line 56 – should be changed to “Association”

Line 62 – I would soften the phrase “not be appropriate”

Participants:

The authors should describe the randomization of sampling. Whether schools were samples according to a randomized method? How were children included?

Procedure:

The authors should explain how diagnoses like spinal pathology were made in order to exclude participants as described in Fig 1? Were they reported by participant? Based on a physician diagnosis?

I would suggest to use the WHO Z-score for calculating overweight and obesity since they are considered the gold standard in the last 8-10 years. But I leave this to the editor to decide. Maybe authors should add a short explanation why they chose this comparison (More precise to developed countries. Etc.)

Line 149 – It was not clear for me whether 8 items questionnaire included only back pain questions of all questions. I suggest to clarify it.

Line 187 – ADL should be mentioned here and how it was assessed.

Results

In general I find the tables to be too loaded. One can't see the forest for the tree. I suggest to reconsider how to present the results.

Table 1: There are some changes that are needed to be done:

BMI column should be deleted. It is clear we are using percentiles.

I would also delete weight and height columns. This is a very loaded table with little interesting information.

Table 2:

Chi square for differences between ages doesn’t really have a clinical meaning. I suggest to split grades into elementary school, junior and high school or calculate P for trend since % is a continuous variable.

Regarding pack pain by weight. Fifty percent in both groups had back pain. Something doesn’t make sense what there is a statistical difference between groups.

Table 3: The most interesting part of the analysis is the ORs. I suggest to show them in the table. Same for table 4.

Line 270 – Where are the ORs and exact P value for the other PB sites? I is important to see the 95%CI.

Lines 289 -291 are not clear. Since OR is not for continuous variables. It is not clear what the association is. I suggest re-phrasing: The risk for... between... to.... was:....

The last paragraph is written unclear and the authors didn’t explain the results correctly. OR is how much the risk of one group compared to the risk is another group for the results. It is not an association. It is not clear which groups were compared in every OR. Why didn’t you adjust for both age and sex together to see the adjusted risk?

I suggest to summarize these results ina table.

Paragraph 3.3 -the same comment as above.

Try and use the above phrase, it will be clearer which groups are compared.

There are some interactions described in the text, please address them (see line 404 in the discussion)

Table 5 – this table doesn’t describe an association.

Lines 354-364 – The authors mention an association but not to which side. For example: “On the other hand, there was an association between ADL restriction and overweight/obese state” Who had lower ADL?

I deeply suggest to write the last paragraph in a different way. Why do the authors repeat chi square test and binary logistic analysis? The reference group for the OR should be mentioned and the group with higher/lower risk should be referred to. This is in concurrence with lack of statistical analysis in Table 6.

Discussion

382-391 belong to the results.

Lines 430-437 repeat the introduction in line 64-82. The authors should interpret possible explanations for the results, in the discussion, rather than in the introduction.

Lines 447-449 contradict line 72-83 in the introduction. While the first give some possible mechanisms, in the discussion it is declared that the possible mechanism is not yet known.  

Paragraph line 468 no numbers are needed.

I am not sure I saw any reference regarding school bags weight. This can be added.

Conclusions:

I suggest to give a sentence of recommendations based on the current results.

 Minor comments:

I am not sure it is correct to write that a logistic regression confirmed the qui square test. The goal is not to compare the 2 statistical tests.

Line 413 – There is one extra “occur”

Author Response

(The authors gave the same response as above.)

Round 2

Reviewer 2 Report

The authors replied all comments and the paper is well written and clear. I hope the comments helped to improve it.